# Effect of Annealing on the Interface and Mechanical Properties of Cu-Al-Cu Laminated Composite Prepared with Cold Rolling

**DOI:** 10.3390/ma13020369

**Published:** 2020-01-13

**Authors:** Xing Fu, Rui Wang, Qingfeng Zhu, Ping Wang, Yubo Zuo

**Affiliations:** 1School of Materials Science and Engineering, Northeastern University, Shenyang 110819, China; fuxingwww@sina.com (X.F.); wwangrrui@163.com (R.W.); 2Key Laboratory of Electromagnetic Processing of Materials, Ministry of Education, Northeastern University, Shenyang 110819, China; zhuqingfeng@epm.neu.edu.cn (Q.Z.); wping@epm.neu.edu.cn (P.W.)

**Keywords:** Cu/Al/Cu laminated composite, interface, intermetallics, tensile strength, elongation

## Abstract

Cu-Al-Cu laminated composite was prepared with cold roll bonding process and annealing was carried out to study the phase evolution during the annealing process and its effect on the mechanical properties of the composite. The experimental results showed that after annealing the brittle intermetallics in the interface mainly includes Al_4_Cu_9_, AlCu and Al_2_Cu. With the increase of annealing temperature, the tensile strength of the composite decreases and the elongation shows a different variation which increases at the beginning and then decreases after a critical point. This phenomenon is related to the evolution of intermetallic compounds in the interface of the composite. It was also found that the crack source of the tensile sample is in the interface and delamination appeared at high annealing temperature (450 °C).

## 1. Introduction

Cu/Al clad laminated composites have been successfully utilized in the fields of electrical and electronic components, coins and decorative materials, due to their advantages associated with high conductivity, low density, good surface performance and price competitiveness over copper and copper alloys [1,2,3]. A Cu/Al clad laminated composite can reduce up to 50% in weight in comparison to equivalent electrical and thermal conductivity of some copper alloys and, the cost can be reduced by 30–50% compared with copper [4]. With both weight reduction and cost saving, Cu/Al clad laminated composite has the ability to replace Cu and Cu alloys in many fields. The research work on the Cu/Al clad laminated composite is still a hot topic and has attracted the interest of many investigators and engineers. Studies on the diffusion bonding, friction-stir welding, hot rolling and cold rolling [3,4,5,6,7] to prepare Cu/Al clad laminated composite have been extensively carried out. Cold rolling has proven to be a successful method to produce Cu/Al clad laminated composites due to the controlled intermetallics layer. Reduction ratio [8] is one of the key parameters for the cold roll bonding process and reduction of 60% of a single pass is usually needed for successful bonding. Asymmetrical cold rolling [9,10] was further developed to produce Cu/Al clad laminated composites with better bonding effect. It was shown that asymmetrical cold rolling provides a remarkable cross shear stress, contributing to the good bonding of the composite. However, a high mismatch speed ratio can reduce the interface bonding strength due to the decrease of rolling force [11]. For the cold rolled Cu/Al clad laminated composites, annealing is needed to soften the material and control the intermetallics layer for the application or further deformation. During the annealing process there will be a brittle Cu_x_Al_y_ intermetallics layer in the interface due to the mutual diffusion of Cu and Al [12]. However, the formation of brittle Cu_x_Al_y_ intermetallic compound at elevated temperature could deteriorate the mechanical and electrical reliability of Cu/Al clad composites. Interfacial structure and properties of Cu/Al clad composites have been studied by many investigators [13,14,15]. Sasaki [16] found that there are continuous intermetallic phases of Al_2_Cu, AlCu and Al_4_Cu_9_ formed at the Cu/Al interface and drawing after annealing results in the formation of ultra-fine copper grains with an approximate diameter of 200 nm near the newly created Cu/Al interface. Hwang et al. [3] and Bellido et al. [12] found that the Cu/Al interface is a three-layer structure including Al_2_Cu, AlCu + Al_3_Cu_4_ and Al_4_Cu_9_. The effect of annealing on the intermetallic phases was also extensively investigated and it was concluded that the thickness of the intermetallics layer increases with increasing annealing temperature, prolonging the annealing time. The mechanical properties of laminated composites was also studied and it was found that the yield strength of the laminated composites follows the law of mixtures [17,18]. However, the effect of interfacial intermetallics on the mechanical properties and further plastic deformation of Cu/Al clad composites fabricated by cold rolling is still under i0nvestigation and the corresponding literature is still limited. In the present work, the Cu/Al/Cu sandwich composites were prepared with cold roll bonding process. The annealing process was carried out to study the evolution of intermetallics during the annealing process, its effect on the mechanical properties and further deformation of the annealed composite.

## 2. Experimental Procedure

Commercial aluminum alloy AA 1060 and pure copper (99.95%) sheets were used as components of the laminated alloys. The initial thicknesses of Cu and Al sheets are 1 mm and 2 mm, respectively. They were cut into specimens with 60 mm in width and 300 mm in length. After surface treatment (brushing and degreasing), Cu-Al-Cu sandwich sheet materials were subjected to cold roll bonding with different reduction ratios ranging between 40% and 80% to prepare Cu/Al/Cu clad sandwich composites. Figure 1a shows the schematic illustration of cold roll bonding process. The as-rolled laminates (with reduction of 80%, final thickness 0.8 mm) were annealed at 200, 250, 300, 325, 350, 400 and 450 °C, respectively, for 60 min in a resistance furnace with Ar protection. Then the annealed composite was further rolled to 0.4 mm with 3 passes. Peel strength of the laminated composite was measured using the peel test [19]. Peel test was performed using SHIMADZU tensile testing machine with a crosshead speed of 5 mm/min. Average peel strength (average load/bond width, N/mm) was used as the peel strength. The dimension of sample for peel test is width 10 mm and length 100 mm. Figure 1b shows the schematic illustration of the peel test.

Microhardness test was performed on the polished Cu-Al-Cu laminated composite by using a micro hardness tester (Cratos W50S, Bright, London, UK) and the load applied during the test was 10g. The average of three tests was used as the microhardness value of the corresponding area. The tensile test samples were selected along the rolling direction and the average of three tests was used as the value of tensile properties for each annealing condition.

The tensile test was carried out using SHIMADZU tensile testing machine (Shimadzu, Kyoto, Japan) with a crosshead speed of 2 mm/min at room temperature. The gage width and gage length for the testing specimen were 6 mm and 25 mm, respectively. SEM (scanning electron microscope), XRD (x-ray diffraction), EDX (energy dispersive x-Ray spectroscopy) and OM (optical microscope) were used to investigate the intermetallics in the interface.

## 3. Results and Discussions

### 3.1. Effect of Reduction on the Peel Strength of Cu-Al-Cu Laminated Composite

The cold roll bonding process was used to prepare Cu-Al-Cu laminated composite. Reduction ratio is one of the key parameters affecting the peel strength. As shown in Figure 2, when the reduction is below 40%, the successful bonding cannot be achieved. In the reduction range of 40 to 50%, bonding of some samples can be achieved although the success ratio is very low. When the reduction is equal or above 60% the successful bonding of the composite can be successfully achieved. When the reduction is below 50% the peel strength is very low, indicating a bad bonding of Cu and Al. With the increase of the reduction, the peel strength keeps increasing and there is a sharp increase between 50% to 60% of the reduction. Based on this, the reduction for the cold roll bonding of Cu-Al-Cu laminated composite should be higher than 60%. Larger reduction can result in better bonding of the composite. It is believed that a higher reduction during cold roll bonding process can provide a flatter asperity that provides a larger contact area between the layers of Cu and Al. In the present work, the highest peel strength of 9.3 N/mm was achieved when the reduction is 80%.

At the reduction of 80%, the composite shows a good bonding without holes or gaps in the interface as shown in OM photo, Figure 3a, and secondary electron image of SEM, Figure 3b. Hereinafter, the Cu-Al-Cu laminated composite prepared by cold roll bonding process with reduction of 80% was used for annealing and further deformation.

### 3.2. Effect of Annealing on the Intermetallics in the Interface

For the cold roll bonded Cu/Al laminated composite, annealing is usually needed to get the controlled properties for the application or further deformation. It is understood that both Al and Cu atoms are thermally activated and the intermetallics layers are formed through diffusion during the annealing process. Figure 4a shows a typical trilaminar intermetallics structure (with the thickness of 17 μm) of the Cu-Al-Cu laminated composite annealed at 450 °C for 60 min. The dotted line represents the linear scan position and the dot represents the position of the spot of EDX analysis. Figure 4b shows the line scanning from layer A(Cu) to layer E(Al) shown in Figure 4a. The both blue vertical dash-lines in Figure 4b indicate the start point and the end point when we calculate the thickness of the intermetallics layer. The intermetallic compound in the interface region is determined by XRD through scanning the peeled surface of the metal. The XRD and line scanning results show that layers A, B, C, D and E are Cu matrices, Al_4_Cu_9_, AlCu, Al_2_Cu and Al, respectively. The layers B, C and D in Figure 4a correspond to rectangular regions B, C and D in Figure 4b. Hwang et al. [3] also found the Al_3_Cu_4_ layer between Al_4_Cu_9_ and AlCu in the Cu/Al laminated composite annealed at 500 °C for 180 min. In the present work Al_3_Cu_4_ layer was not found, which could be due to the lower annealing temperature (450 °C and below) and short annealing time (60 min). As shown in Figure 4a, the thicknesses of the layers follow the sequence of Al_2_Cu > Al_4_Cu_9_ > AlCu. As the diffusion of Cu in Al is faster than that of Al in Cu [3,20], Al_2_Cu is presumed to be formed more quickly than the phase of Al_4_Cu_9_. Therefore, the layer of Al_2_Cu is thicker than Al_4_Cu_9_. After the formation of Al_2_Cu and Al_4_Cu_9_, AlCu forms in between. Figure 4d shows the microhardness of different intermetallics in the interface of Cu and Al. Combining Figure 4b, the hardness of the Cu matrix, Al_4_Cu_9_, AlCu, Al_2_Cu, and Al matrix is 60.8 ± 8.4, 595.4 ± 50.4, 432.8 ± 30.8, 208.4 ± 20.5 and 19.5 ± 2.5 HV, respectively. It can be seen that the intermetallics of the interface is much harder than the Cu matrix and the Al matrix. 

Figure 5 shows the interface development of Cu/Al laminated composite with the increase of annealing temperature. The variation of the thickness of intermetallics layers as a function of annealing temperature is shown in Figure 6. For the as-bonded composite, no obvious interface development can be observed, as shown in Figure 3. For the annealed composite, when the annealing temperature is 250 °C, an interfacial layer with a thickness about 1.0 μm developed, as shown in Figure 5a. As the annealing temperature increases to 300, 350, 400 and 450 °C, the thickness of the interfacial layer increases to 2.4, 4.4, 9.2, and 17 μm, respectively, as shown in Figure 5b–d and Figure 4b. It needs to be noted that we measured the thickness in three spots and did linear scanning with EDS at one spot to confirm the thickness of the intermetallics layer. The value of the thickness is based on the line scanning results, as shown in Figure 4b and Figure 5. 

It is clear that under annealing conditions, the interface is a three-layer structure and the thickness of the intermetallics layer increases with the increase of annealing temperature. As shown by the fitted curve in Figure 6, the thickness of the intermetallics layer increases following an exponential relationship in the temperature range between 200 °C (473 K) and 450 °C (723 K). The relationship is mainly attributed to increased thermal energy of Cu and Al atoms at higher temperature [21]. With the increase of annealing temperature, the Cu and Al atoms can obtain more thermal energy to break through the barrier of diffusion energy, which can achieve a wide range of free migration and accelerate the growth of the diffusion layer. The relationship between the growth thickness *D* of the intermetallic compound and the heat treatment temperature *T* obeys the Arrhenius equation. According to the data in the text, it can be specifically described as Equation (1):(1)$$D=0.47541+0.0007674e^{\left(\frac{T}{72.464}\right)},$$
where *D* is the thickness of the intermetallics layer and *T* is the annealing temperature (K).

### 3.3. Effect of Annealing on the Mechanical Properties

Figure 7 shows the stress–strain curves of the composites annealed at different temperatures and Figure 8 shows the variation of the strength and elongation as a function of annealing temperature. For the as-bonded composite, the tensile strength and elongation are 270 MPa and 5.5%, respectively. For the annealed composite, with the increase of annealing temperature, the tensile strength decreases slowly at the beginning, and then quickly, and there is a sharp decrease from 220 MPa to 164 MPa in the annealing temperature range of 300–325 °C. After 325 °C, further increasing the annealing temperature just leads to a slight decrease of the tensile strength. In the case of elongation, when the composite is annealed at 200 °C, the elongation shows a slight decrease compared with that of the as-bonded composite. With increasing the annealing temperature in the range of 200–350 °C the elongation keeps increasing and there is a sharp increase from 11.5% to 26% when the annealing temperature increases from 300 °C to 325 °C. The peak of the elongation, 32.5%, appears at 350 °C. However, with further increasing annealing temperature, the elongation begins to decrease. With regard to elongation, the proper annealing temperature range is from 325–400 °C and at 350 °C the highest elongation was obtained in the present work.

It is understood that the decrease of tensile strength and the increase of elongation with increasing the annealing temperature is due to the recovery and recrystallization during the annealing process. The decrease of the elongation when the annealing temperature is over 350 °C could be related to the evolution of intermetallics in the Cu/Al interface. In order to clearly understand the mechanism, the variation of intermetallics in the interface close to the fracture plane of the tensile samples was observed in secondary electron image of SEM. The results are shown in Figure 9.

It can be seen there are some cracks in the interface of the sample after tensile test and the size of the cracks increases with the increase of annealing temperature. Due to the high brittleness of intermetallics, the cracks initiates at the intermetallics layer during stretching process. Once the intermetallics layer is broken, the broken intermetallics keep being bonded on the matrix metal and stress concentration is generated near the matrix metal surface in the crack gap. The necking therefore appears in the matrix metal of Cu and Al, as clearly shown by the arrows in Figure 9d,e. With a high annealing temperature (450 °C), serious delamination was also observed. Some composite samples even separates to three individual layers after tensile test. Figure 9e,f gives a typical delamination structure of the stretched tensile sample annealed at 450 °C. 

Lower ductility of intermetallics than that of the Al and Cu matrix is believed to be one of the main reasons for delamination. EDX results showed that layers 1, 2 and 3 in Figure 9f are Al_2_Cu, AlCu and Al_4_Cu_9_, respectively. Interfacial delamination usually initiates in the CuAl intermetallics layer and mostly in the interface of Al_2_Cu and AlCu. As reported by Ferreira [22], Cu-rich intermetallics have a smaller atomic volume and will generate a volumetric shrinkage during the formation of these phases. Higher annealing temperature leads to the formation of a higher volume of Cu-rich intermetallics and consequently larger volumetric shrinkage. This is expected to cause a large internal stress during the annealing process and consequently contribute to the formation of delamination. The difference of the ductility of different intermetallics could also contribute to the formation of delamination. When the annealing temperature is 400 °C no obvious delamination is observed, although there are some micro-delaminations (interlayer crack) as indicated by the circle in Figure 9d. When the annealing temperature is 350 °C and below, there is no delamination.

At higher annealing temperatures, the brittle intermetallics layer becomes thicker. During the stretching process, brittle intermetallics are broken first and then cracks form. With further stretching, the necking of the matrix metal in the crack occurs. This could be the main reason for the reduced elongation at high annealing temperatures. Furthermore, delamination appeared at a high annealing temperature (450 °C). Due to the delamination, the constraint of the interlayer disappears, which is also believed to contribute to the decrease of elongation. In addition, when the annealing temperature is high, the elongation of copper and aluminum can be mismatched. The shear stress, due to mismatch elongation, can causes further damage to the interface during the tensile process.

### 3.4. Evolution of Intermetallics after Further Plastic Deformation

As discussed in Section 3.3, the thick intermetallics layer can lead to the delamination and the decrease of elongation, which is unwanted for further cold rolling. It is therefore important to investigate the effect of intermetallics on the further cold rolling. The cold rolling with total reduction of 50% was carried out to the annealed composite and the intermetallics after further cold rolling was observed. As shown in Figure 10, with further cold rolling, the intermetallics are broken and are still distribute along the interface. The newly formed interface can also be clearly observed. For the sample annealed at 450 °C, after further cold rolling, some of the broken intermetallics rotate at an angle as indicated by the arrows in Figure 10c. No obvious delamination and voids were observed, although the intermetallics are broken during the cold rolling process. For the samples annealed at 350 °C and 400 °C, after further cold rolling the composite shows a similar behavior, but a much thinner thickness and smaller size of broken intermetallics is present.

## 4. Conclusions

Cu-Al-Cu laminated composite was prepared with cold rolling process and annealing was carried out to study the phase evolution during the annealing process and its effect on the mechanical properties of the composite. After annealing the brittle intermetallics in the interface mainly include Al_4_Cu_9_, AlCu and Al_2_Cu. With the increase of the annealing temperature, the thickness of the internetallics layer increases following an exponential relationship in the temperature range between 200 °C (473 K) and 450 °C (723 K). With the increase of the annealing temperature, the strength keeps decreasing, while the elongation shows an increasing trend at the beginning and then a decreasing trend after the peak of 32.5% (annealed at 350 °C). Large thickness of brittle intermetallics layer, cracks of the brittle intermetallics during stretching and necking of the matrix metal in the crack gap are believed to be the main reasons for the reduction of elongation at high annealing temperatures. In addition, delamination appeared at a high annealing temperature (450 °C), which is also believed to contribute to the decrease of elongation. With further cold rolling of the annealed composite, the intermetallics are broken, and a new bonding interface forms without the formation of delamination and void.

## Figures and Tables

**Figure 1 materials-13-00369-f001:**
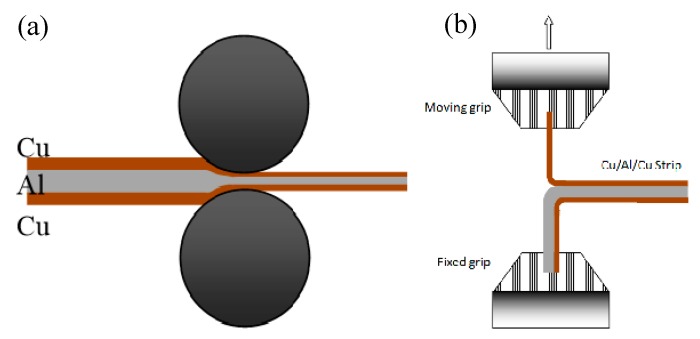
Schematic illustration of the cold roll bonding of sandwich composites (**a**) and the peel test (**b**).

**Figure 2 materials-13-00369-f002:**
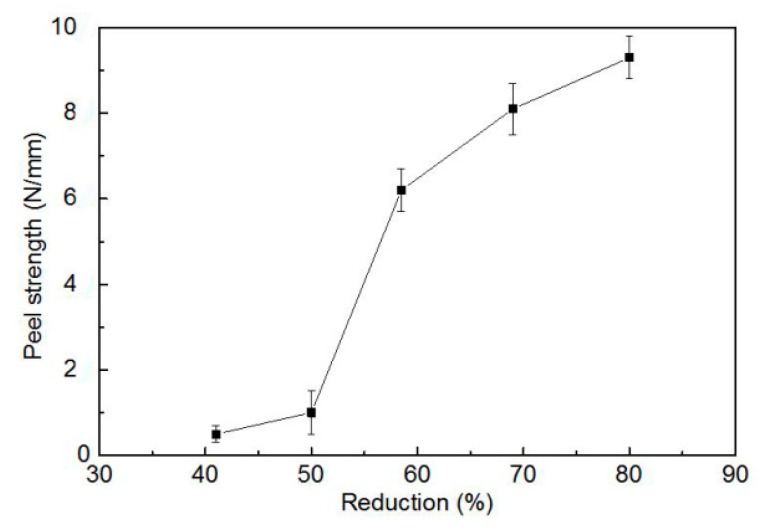
Variation of peel strength as a function of rolling reduction.

**Figure 3 materials-13-00369-f003:**
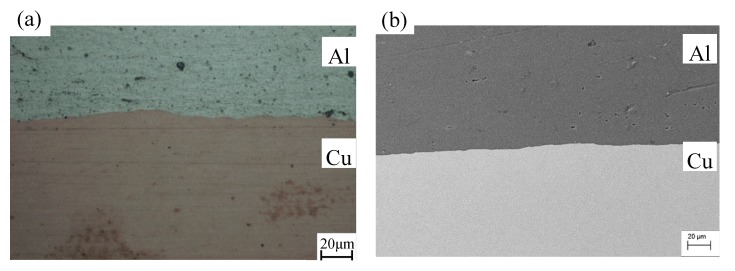
The structure of Cu/Al laminated composite prepared by cold roll bonding with the reduction of 80%: (**a**) OM photo; (**b**) secondary electron image of SEM.

**Figure 4 materials-13-00369-f004:**
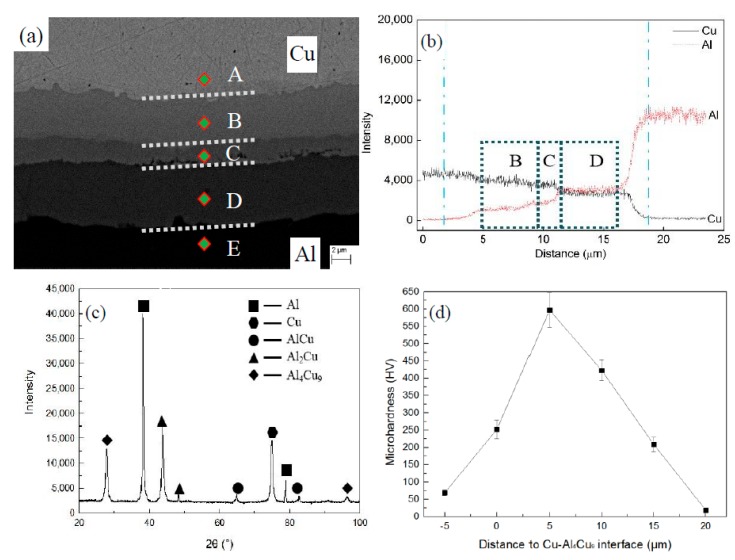
Intermetallics in the interface of Cu and Al: (**a**) backscattered electron image of SEM; (**b**) line scanning with EDX; (**c**) XRD measurement of peeled surfaces of component metals; (**d**) microhardness of different intermetallics.

**Figure 5 materials-13-00369-f005:**
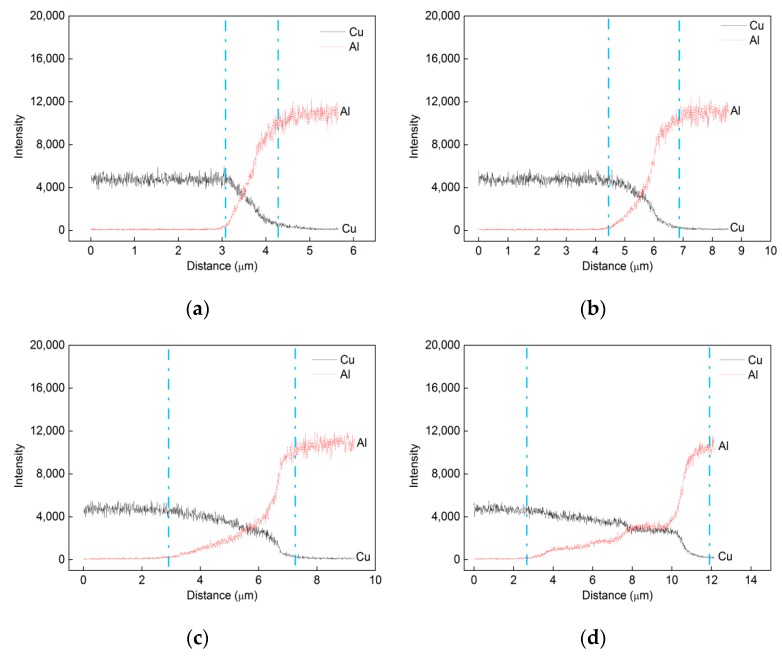
Interface development of Cu/Al laminated composites annealed under various temperatures: (**a**) 250 °C; (**b**) 300 °C; (**c**) 350 °C; (**d**) 400 °C.

**Figure 6 materials-13-00369-f006:**
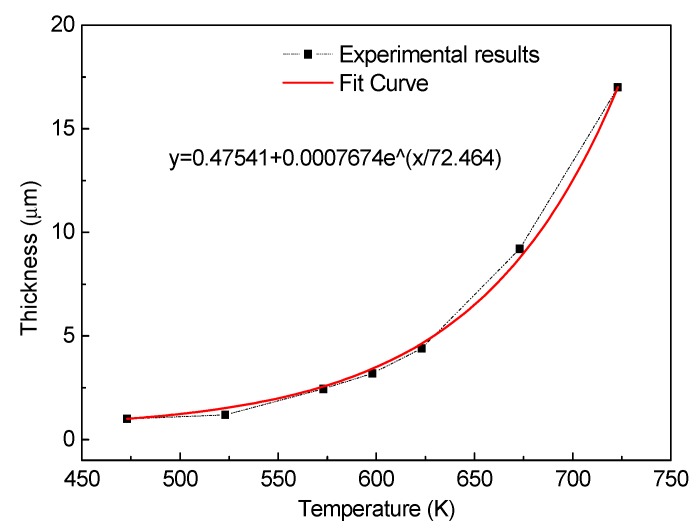
Variation of the thickness of intermetallics layer as a function of annealing temperature.

**Figure 7 materials-13-00369-f007:**
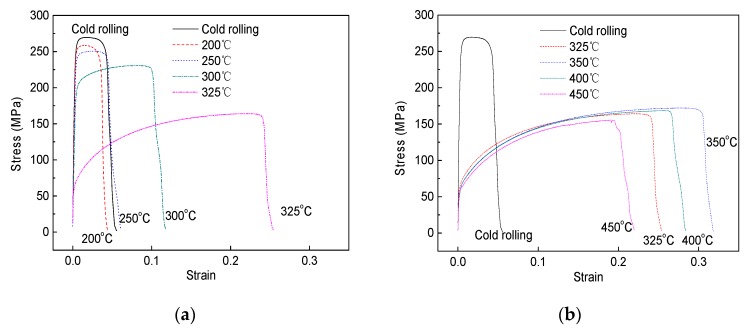
Stress-strain curves of the composites annealed at different temperatures: (**a**) 200–325 °C; (**b**) 325–450 °C.

**Figure 8 materials-13-00369-f008:**
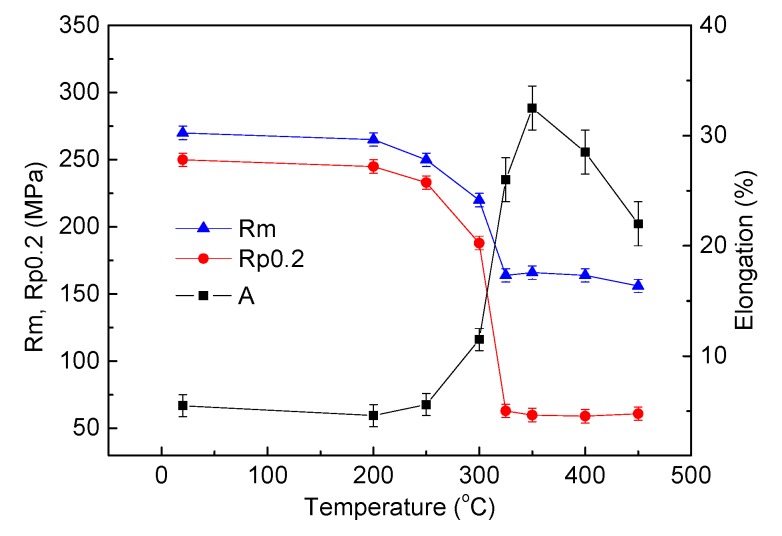
Variation of the strength and elongation as a function of annealing temperature.

**Figure 9 materials-13-00369-f009:**
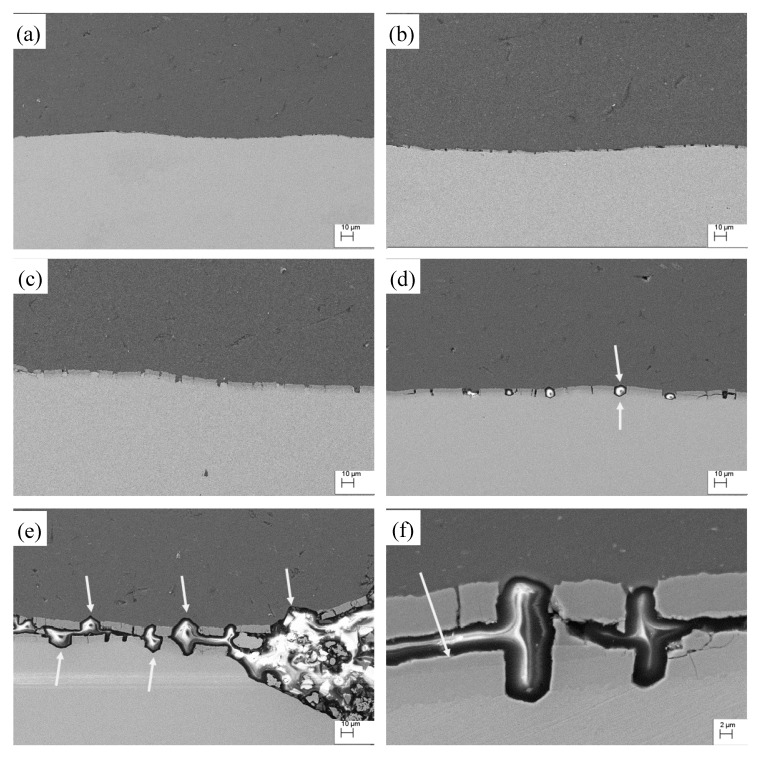
Variation of the intermetallics close to the fracture plane after tensile test. The tensile samples were annealed at (**a**) 300 °C; (**b**) 325 °C; (**c**) 350 °C; (**d**) 400 °C; (**e**) and (**f**) 450 °C.

**Figure 10 materials-13-00369-f010:**
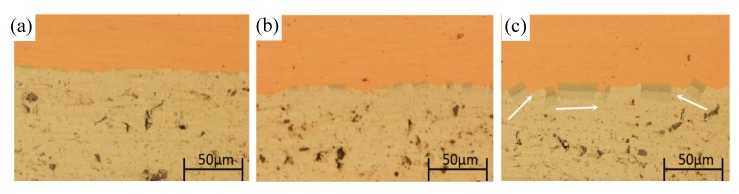
Evolution of intermetallics with further cold rolling. Before cold rolling the composites were annealed at (**a**) 350 °C; (**b**) 400 °C; (**c**) 450 °C.

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
