# Peer review of "Effect of Annealing on the Interface and Mechanical Properties of Cu-Al-Cu Laminated Composite Prepared with Cold Rolling"

_materials, 2020, doi:10.3390/ma13020369_

Round 1

Reviewer 1 Report

The manuscript studies preparation of Al/Cu laminated composite using room temperature roll bonding. The prepared composites are subjected to annealing heat treatment to study for changes in the interface microstructure and mechanical properties. The manuscript lacks in several critical aspects and needs further data and analysis to support the research findings. Following are comments on the manuscript:

The composite interface is analyzed using EDX and are presented in Figure 4. Why not perform XRD in addition to confirm the intermetallics present at the interface. Also, the mechanical property of composite is governed by the interface region and after annealing there is change in the interface microstructure. Since the interface is critical, the change of mechanical property should also be studied locally using technique such as micro- and/or nano-indentation measurements to understand how the intermetallics affect the overall composite property. Without this data and only microscopy images it is difficult to ascertain the composite mechanical property to the intermetallics present at interface. How was the tensile testing done – along or perpendicular to the rolling direction? Mention it explicitly in the methods section. Include how many samples were tested for tensile property.

    3. Figure 5 –  image quality is not up to journal standards with so much            scratches.

Reviewer 2 Report

l. 86: Nothing about the reduction under 40 % can be stated from Fig. 2

Use always the same order of the layers - Cu top, Al bottom.

Why do you use sometimes SE and sometimes BSE signal in SEM? Use only one of them. Which signal did you use for Fig. 9?

Errase the information bar from the SEM images and insert your own scalebar.

l. 118: form => formed

l. 121: The last sentence is redundant

Fig. 4 - What do the vertical dash-and-dot lines represent?

l. 131: decreases => increases

l. 131: In how many spots did you measure the thickness of the layer

l. 139: What does the "improved thermal energy" mean?

Can you compare the values in Eq. 1 with some other work/theoretical background?

l. 162: However, WITH further

Fig. 9: What are the dark spots in the upper part of images b) and c)?

l. 210: leads => lead

Fig. 10 - The scalebar is not readable. Better use image of only one Cu-Al interface to show the broken intermetalics layer in better detail. 

Reviewer 3 Report

The paper is a good technical report. Its scientific importance is, however, quite limited. Nevertheless, it is rather well written (in very comprehensive way) so I recommend the paper for publication after corrections (which are needed).

Some points have to be explained and corrected:

The fig. 5 is of poor quality – many scratches are present and different magnification is used, which makes impossible to compare the layer thicknesses.

In fig. 9b and c, some porosity can be seen. Is it due to the proximity of fracture surface or the porosity was generated during annealing? This have to be more elaborated and explained in the text. It is also connected with previous remark on fig. 5 (the porosity, if present, should already be seen on properly prepared sample surface).

Micrographs of fracture surfaces should be shown, especially in the interface areas where the crack initiation (according to authors) occurs. Otherwise the statements are hypothetic only.

The discussion on effect of intermetallics on further cold rolling is not so clear, because no tensile tests are provided. Compressive deformation could significantly decrease bonding of the (broken) intermetallic particles to the matrix (with no apparent signs). This should be verified more properly.

Other remarks:

All abbreviations (EDX, SEM, OM …) should be explained at the first appearance.

Please, use standard SI unit (N/mm instead N/cm etc.). Please choose if you want to use degrees of centigrade or Kelvin units (throughout the text)

In equations, please, use correct variables (e.g. T for temperature) not x and y.

Reviewer 4 Report

The idea of paper is no so bad, but the work is confused with many methodological errors and many imprecisions. For example fig.3, fig.4, fig.9 are confused. The authors don't declare the load of micro hardness test and the indentation are too big to evaluate with precision the hardness of some layer. The results.of hardness test don't have the standard deviation. The authors affirm that the high hardness is due at the presence of some precipitate, but only the Edx analysis is no enough to identify the type of precipitate because is only a qualitative chemical analysis. The paper appears like an internal draft and not a paper ready for a submission at a journal. I suggest to reject this paper.

Round 2

Reviewer 1 Report

Regarding the micro/nano hardness measurements the authors have responded "It is known that the aluminum-copper compounds are hard and brittle, based on literatures and it is believed that the brittleness of the compounds are the key factor affecting the plasticity of the composite materials. However the hardness test cannot give the plasticity information. This is why we did not do the hardness test. "

The interface is affected by the heat treatment and governs the mechanical property of the composite. The reply from the authors is contradicting. The micro/nano hardness helps in quantifying the change in the hardness (brittle/ductile) across the interface which in turn can offer insight on the overall composite mechanical properties. Without this data, it is difficult to correlate the composite mechanical property to the interface based only on EDS map.

Reviewer 2 Report

Dear Authors,

you did answer most of my questins in the response letter, however, most of the changes did not appear in the new manuscript I got.

See the attachement for details.

Reviewer 4 Report

I'm not totally satisfied about the revisions. The revisions are not sufficient. The authors don't explain in which way they can affirm that we havethe presence of some specific precipitate only trough EDX. Please explain better this point thath is the key poiny of the paper.
Please in line 131 insert the standard deviation.

Author Response

Thank you so much for your comments.

Please see the revised paper.